# Effects of Primary and Secondary Psychopathy on Deontological and Utilitarian Response Tendencies: The Mediator Role of Alexithymia

**DOI:** 10.3390/healthcare10091650

**Published:** 2022-08-29

**Authors:** Zhihui Wu, Xiyou Chen, Daoqun Ding, Shengqi Zou, Shenglan Li, Xiangyi Zhang

**Affiliations:** 1Department of Psychology, School of Education Science, Hunan Normal University, Changsha 410081, China; 2Cognition and Human Behavior Key Laboratory of Hunan Province, Hunan Normal University, Changsha 410081, China; 3Center for Mind and Brain Science, Hunan Normal University, Changsha 410081, China; 4Normal College, Hunan University of Arts and Science, Changde 415000, China

**Keywords:** primary psychopathy, secondary psychopathy, deontological response tendencies, utilitarian response tendencies, alexithymia

## Abstract

(1) Background: The purpose of this study is to provide more nuanced insights into the effects of sub-dimensional levels of psychopathy on moral dilemma judgments. To this end, this study examined the effects of primary and secondary psychopathy on utilitarian and deontological response tendencies. Moreover, this study also explored the mediating role of alexithymia as well as the moderating role of gender in these effects. (2) Methods: A total of 1227 participants were recruited through the online questionnaire service wjx.cn. After deleting unfinished questionnaires, the remaining 1170 participants were included in the final data analysis. Each participant completed a demographic information questionnaire, the Levenson Self-Report Psychopathy Scale, the Toronto Alexithymia Scale-20, and six pairs of moral dilemmas. Descriptive and correlational analyses of study variables were conducted in SPSS 22.0. Mediation and gender difference analyses were conducted in AMOS 23.0. (3) Results: Primary psychopathy was negatively correlated with deontological response tendencies and uncorrelated with utilitarian response tendencies. By contrast, secondary psychopathy also correlated negatively with deontological response tendencies, but it correlated positively with utilitarian response tendencies. Mediation analysis revealed that alexithymia only mediated the relationship between secondary psychopathy and deontological response tendencies. Multi-group analysis revealed that there was no difference between females and males in the indirect effect model. (4) Conclusions: People with high primary psychopathy are less likely to reject harm in moral dilemmas. By contrast, people with high secondary psychopathy have high alexithymia, which causes them to be less concerned about avoiding harm, and they are more likely to maximize outcomes in moral dilemmas. These findings shed new light on the moral dilemma judgments of individuals with primary and secondary psychopathy.

## 1. Introduction

Psychopathy is characterized by callousness, a lack of empathy and remorse, profound egocentricity, and antisocial behavior [1]. Psychopathy and antisocial personality disorder overlap in some respects, but the interpersonal and affective deficits in psychopathy are more obvious [2,3]. For example, individuals with high psychopathy have extensive emotional impairments, as well as deficits in various empathic processes and the perception of one’s own emotions [4]. Moreover, the structure of psychopathy is multidimensional in nature [1,5]. Based on the two-factor model of psychopathy, psychopathy can be divided into primary psychopathy and secondary psychopathy [5,6]. Primary psychopathy is thought to be a result of biological deficits, which is characterized by interpersonal and affective deficits as well as a low level of anxiety [5,7,8]. Secondary psychopathy is thought to be a result of social disadvantages, which is characterized by antisocial and general behavior problems as well as a high level of anxiety [5,7,8]. The four-factor model of psychopathy comprises interpersonal (e.g., pathological lying), affective (e.g., lack of remorse or guilt), lifestyle (e.g., impulsivity), and antisocial facets (poor behavior controls) [1,9]. These four correlated facets can be further summarized by two higher-order factors: Factor 1 (Inter-personal/Affective) and Factor 2 (Social Deviance). Although the two-factor model of psychopathy differs in some respects from the four-factor model, numerous studies support that primary psychopathy and secondary psychopathy are considered to essentially define the same structures as Factor 1 and Factor 2 [6,10,11].

### 1.1. Moral Dilemma Judgment

Research on moral dilemma judgment has formed the view that utilitarian judgment and deontological judgment can be measured by using sacrificial moral dilemmas [12]. For example, in the crying baby dilemma, you and some of your townspeople will be found and killed by enemy soldiers if you do not smother the crying baby [13]. The participants are asked whether they would sacrifice the crying baby to save themselves and the others according to their personal opinions. From a utilitarian view, smothering the baby would be morally acceptable, because such an action could maximize overall well-being (i.e., sacrificing the baby could save oneself and others). In contrast, from a deontological view, smothering the baby would be morally unacceptable, because such harmful action violates moral principles. Thus, accepting harm in similar moral dilemmas would be described as utilitarian judgment, while rejecting harm would be described as deontological judgment [14].

According to dual process theory, more powerful cognitive deliberations seem to promote utilitarian judgment [13,15,16]. Patil et al. (2021) demonstrated a positive relationship between reasoning ability and utilitarian judgment [17]. By contrast, more powerful affective processes promote deontological judgment [13,15,16]. For example, considerable studies have found that individuals who score high on measures of affective processing tend to make deontological judgments in moral dilemmas, such as outcome aversion [18] and empathic concern [19]. However, several studies found that clinical patients who exhibit deficits in affective processing show a preference for utilitarian judgment in moral dilemmas, such as those with lesions in the ventromedial prefrontal cortex [20]. Moreover, nonclinical individuals with high psychopathy also show a preference for utilitarian judgment in similar moral dilemmas [18,21].

### 1.2. Psychopathy and Moral Dilemma Judgment

The relationship between psychopathy and moral dilemma judgment has received considerable attention from clinical and moral psychology researchers. Such empirical evidence not only helps to explain other main aspects of psychopathy but also provides a deeper understanding of the mental underpinnings of moral judgment [12]. Using sacrificial dilemmas similar to the crying baby, some studies have shown a positive correlation between psychopathy and utilitarian judgment, suggesting that people with high psychopathy exhibit a preference for utilitarian over deontological judgment [12,18,19,21,22,23,24,25,26,27,28]. However, the majority of these studies treated psychopathy as a general, unitary construct, which may obscure differential associations between the sub-dimensional levels of psychopathy and moral dilemma judgment. Even though several studies have investigated the effects of sub-dimensional levels of psychopathy on moral dilemma judgment, such effects remain complex and elusive. For instance, some studies found that only primary psychopathy was correlated with utilitarian judgment [24,28], whereas other studies found that only secondary psychopathy was related to utilitarian judgment [22,26]. It has also been found that primary and secondary psychopathy were correlated with utilitarian judgment [25]. Thus, the effects of sub-dimensional levels of psychopathy on moral dilemma judgment need to be further explored.

More importantly, the majority of studies on the moral judgment of psychopathy adopted the traditional moral dilemma paradigm. The issue with the traditional moral dilemma paradigm is that deontological judgment is treated as the pure inverse of utilitarian judgment (i.e., making fewer deontological judgments implies more utilitarian judgments and vice versa). The response to moral dilemmas only reflects a relative preference for one over the other rather than absolute preferences for either. This leads the psychological processes underlying utilitarian and deontological judgment to have an inverse relationship, in which stronger utilitarian response tendencies signify weaker deontological response tendencies and vice versa [29]. However, numerous studies support that the processes underlying utilitarian and deontological judgment are distinct and independent [14,29,30,31]. For example, individuals with high alexithymia showed a preference for utilitarian judgment due to a reduction in deontological response tendencies, whereas utilitarian response tendencies were not affected [31].

Using a novel process dissociation (PD) approach could independently quantify the strength of utilitarian and deontological response tendencies (for the details of the PD approach, see the Section 2) [29]. Utilitarian response tendencies reflect the notion that people always aim to maximize good consequences regardless of whether doing so entails causing harm. By contrast, deontological response tendencies reflect the notion that people always aim to reject causing harm regardless of whether doing so maximizes consequences or not [29,32]. A meta-analysis found that the utilitarian and deontological response tendencies had robust correlations with traditional moral dilemma judgment (i.e., treating deontological judgment as the pure inverse of utilitarian judgment), but these two response tendencies were only mildly correlated (r = 0.1) or uncorrelated, which demonstrated that the utilitarian and deontological response tendencies were independent contributions to moral dilemma judgment [30]. 

### 1.3. Current Study

The purpose of this study is to provide more nuanced insights into the effects of sub-dimensional levels of psychopathy on moral dilemma judgments. To this end, this study used a novel process dissociation (PD) approach to independently quantify the strength of utilitarian and deontological response tendencies, and examined the effects of primary and secondary psychopathy on utilitarian and deontological response tendencies. According to dual process theory, deontological response tendencies involve relatively more affective responses to harmful actions, whereas utilitarian response tendencies involve a relatively more deliberative reasoning about outcomes [15,29,32]. Previous studies have shown that primary psychopathy is characterized by interpersonal and affective deficits [7,8]; these facets have been linked to lower empathic concern [33]. Given that empathic concern uniquely predicts deontological response tendencies, rather than utilitarian response tendencies [18,29], this study hypothesized that primary psychopathy would show a significant negative correlation with deontological response tendencies. By contrast, secondary psychopathy is characterized by antisociality and general behavior problems [7,8]. These characteristics have been uniquely associated with impulsivity, violence, and physical aggression, which may blunt emotional reactions for causing harm and decrease overall well-being [1,34]. Thus, this study hypothesized that secondary psychopathy would show a significant negative correlation with deontological and utilitarian response tendencies.

Moreover, the mechanism of the relationship between primary and secondary psychopathy and deontological and utilitarian response tendencies requires further investigation. Psychopathy is characterized by a lack of emotional awareness and shallow emotions [1,4]. A concept associated with reduced emotional awareness and emotionality is alexithymia [35]. Alexithymia is a multidimensional personality trait characterized by difficulty in identifying and describing feelings, as well as by an externally oriented thinking style [36]. When individuals with high alexithymia feel upset, they cannot determine whether it is sadness, fright or anger [37]. Patil (2015) found that individuals with high psychopathy show a preference for utilitarian judgment because they have less aversion toward performing harmful actions [21]. Reduced action aversion was, in turn, attributed to an impaired understanding of one’s own emotions. These findings give rise to the assumption that alexithymia plays a causal function in the relationship between psychopathy and moral judgment. A meta-analysis showed a positive relationship between psychopathy and alexithymia (r = 0.21), as well as a positive relationship between primary and secondary psychopathy and alexithymia [4]. Moreover, alexithymia is uniquely associated with deontological response tendencies, rather than utilitarian response tendencies [31]. Therefore, the current study hypothesized that alexithymia would mediate the relationship between primary and secondary psychopathy and deontological response tendencies.

Furthermore, evidence from previous studies suggests that there are gender differences in psychopathy, alexithymia, and moral judgment [38,39,40,41]. For example, men scored higher than women on total psychopathy, primary psychopathy, and secondary psychopathy [39,42]. In both clinical and nonclinical populations, men exhibit higher levels of alexithymia compared to women [40,41]. However, women score higher than men on deontological tendencies in moral dilemma judgments [38]. Therefore, whether there are gender differences in these relationships needs to be further tested.

## 2. Materials and Methods

### 2.1. Participants

A total of 1227 participants were recruited through the online questionnaire service wjx.cn. After deleting unfinished questionnaires, the remaining 1170 participants (703 females, 467 males, age range 17–24 years, *M*_age_ = 18.86 years, *SD*_age_ = 1.12, the number of participants in each age group (see the Appendix A) were included in the final data analysis. A power analysis conducted in G*power (Version 3.1.9.2) indicated that the final sample of 1170 participants provided a power of 95% in detecting a correlation with a small effect size of ρ = 0.10 (two-tailed) [43]. This study was approved by the Research Ethics Committee of Hunan Normal University. Electronic informed consent was obtained from all participants. After the participants submitted the questionnaires, they received CNY 10 compensation.

### 2.2. Measures

#### 2.2.1. Primary and Secondary Psychopathy

The Levenson Self-Report Psychopathy Scale (LSRP) was used to measure individuals’ level of psychopathy, which is a self-report measure with sound reliability and validity [5,44]. The LSRP was divided into 2 subscales as follows: primary psychopathy was designed to assess the interpersonal and affective features of psychopathy (e.g., I enjoy manipulating other people’s feelings); secondary psychopathy was designed to assess impulsivity and other antisocial behaviors (e.g., I find myself in the same kinds of trouble, time after time). The LSRP comprises 26 items, and each item was coded from 1 (strongly disagree) to 4 (strongly agree). Higher total scores reflect individuals’ higher level of psychopathy. The Chinese version of the LSRP is reliable and valid in the Chinese population [45]. Cronbach’s alpha was 0.85 for the LSRP, 0.86 for primary psychopathy and 0.71 for secondary psychopathy in the current study.

#### 2.2.2. Alexithymia

The Toronto Alexithymia Scale-20 (TAS-20) was used to measure individuals’ level of alexithymia, which is a self-report measure with well-demonstrated reliability and validity [36]. The TAS-20 was divided into 3 subscales as follows: difficulty in identifying feelings; difficulty in describing feelings; and externally oriented thinking. The TAS-20 comprises 20 items, and each item was coded from 1 (strongly disagree) to 5 (strongly agree). Higher total scores reflect individuals’ higher level of alexithymia. The Chinese version of the TAS-20 is reliable and valid in the Chinese population [46]. Cronbach’s alpha was 0.80 in the current study. 

#### 2.2.3. Utilitarian and Deontological Response Tendencies

Using a process dissociation (PD) approach could independently quantify the strength of utilitarian response tendencies and deontological response tendencies. The key to PD analyses is to employ both incongruent and congruent moral dilemmas [29]. Participants completed 6 incongruent and 6 congruent moral dilemmas, which were selected from Conway and Gawronski (2013) (for the Chinese version of the incongruent and congruent moral dilemmas see the Appendix A). 

Incongruent dilemmas approximate the traditional high-conflict dilemma, in which harmful actions promote the greater good. Utilitarian and deontological response tendencies would drive divergent responses in incongruent dilemmas. For instance, in the incongruent dilemma of the crying baby, participants were asked whether it was appropriate to smother the child to prevent themselves and other townspeople from being killed. In this case, utilitarian response tendencies would lead people to agree to sacrificing the baby because sacrificing one person would save more lives. In contrast, deontological response tendencies would drive people to disagree with the baby sacrifice because such harmful behavior violates moral norms. Congruent dilemmas have a similar structure and wording to incongruent dilemmas, except for the outcomes: endorsing harmful actions would lead to worse outcomes overall [29]. Thus, utilitarian and deontological response tendencies would lead to identical responses in congruent dilemmas. For example, in the congruent dilemma of the crying baby, participants were asked whether it was appropriate to smother the child to prevent themselves and other townspeople from being captured to work in the quarries. In this case, utilitarian and deontological response tendencies would drive people to disagree with the baby sacrifice because dealing harm would lead to worse outcomes overall. 

In other words, utilitarian response tendencies always drive people to reject harm in congruent dilemmas and to accept harm in incongruent dilemmas because action can maximize good outcomes, whereas deontological response tendencies always drive people to reject harm in incongruent and congruent dilemmas because action would violate moral norms. 

Utilitarian response tendencies (*U* parameter) were obtained from the difference in the proportion of “unacceptable” responses between congruent and incongruent dilemmas [29]:*U* = *p* (unacceptable|congruent) − *p* (unacceptable|incongruent).

A higher *U* parameter demonstrates that participants tend to reject harmful actions when they fail to maximize good outcomes (i.e., congruent dilemmas) and accept harmful actions when they could maximize good outcomes (i.e., incongruent dilemmas). Scores of *U* range from −1 to 1.

Deontological response tendencies (*D* parameter) were obtained from the proportion of “unacceptable” responses in incongruent dilemmas relative to all nonutilitarian responses:*D* = *p* (unacceptable|incongruent)/(1 − *U*).

A higher *D* parameter demonstrates that participants tend to reject causing harm regardless of whether doing so maximizes consequences or not. Scores of *D* range from 0 to 1.

### 2.3. Procedure

Participants completed all measures online. At the beginning, participants were required to read the instructions. Then, participants completed the electronic informed consent and demographic information questionnaire. Subsequently, they completed the LSRP and TAS-20. Finally, participants completed 6 incongruent and 6 congruent moral dilemmas. For each dilemma, participants were asked to indicate whether the described action would be appropriate or inappropriate (“yes, it was appropriate”, or “no, it was inappropriate”) according to their personal opinion.

### 2.4. Data Analysis

Descriptive and correlational analyses of study variables were conducted in SPSS 22.0. Structural equation modeling (SEM) was conducted in AMOS 23.0, which was used for testing the indirect effect model and its gender differences. Firstly, a measurement model was built to test whether indicators could well-represent relevant latent variables. Using the technique of item-to-construct balance [47], this study built three parcels for primary psychopathy, secondary psychopathy, and alexithymia, in order to reduce measurement errors [48]. The fit of the model was acceptable if the root mean square error of approximation (RMSEA) and standardized root mean square residual (SRMR) values were below 0.08, and comparative fit index (CFI) and Tucker–Lewis index (TLI) values were above 0.90. [49]. The model fit the data well (*χ*^2^/*df* = 5.34, RMSEA = 0.06, CFI = 0.97, TLI = 0.96, SRMR = 0.05), and all the factor loadings were highly significant (*p* < 0.001). Secondly, maximum likelihood estimation (ML) with bootstrapping (with 5000 replicates and a 95% confidence interval) was used to evaluate whether the indirect effects were significant in the structural model with age and gender as control variables. Finally, a multi-group analysis was conducted to determine whether females and males differed in the indirect effect model.

## 3. Results

Overall, participants judged that harmful actions were more acceptable in incongruent dilemmas (*M* = 46%, *SD* = 23%) than in congruent dilemmas (*M* = 32%, *SD* = 22%), *t* (1169) = 20.33, *p* < 0.001. For descriptive statistics for each dilemma, see the Appendix A.

### 3.1. Preliminary Analysis

The descriptive and Pearson correlations among the study variables are shown in Table 1. As expected, primary psychopathy was negatively correlated with the *D* parameter (i.e., deontological response tendencies) and uncorrelated with the *U* parameter (i.e., utilitarian response tendencies). Secondary psychopathy also correlated negatively with the *D* parameter. However, contrary to the hypothesis, secondary psychopathy correlated positively with the *U* parameter. Moreover, primary and secondary psychopathy correlated positively with alexithymia, but alexithymia was only negatively correlated with the *D* parameter and uncorrelated with the *U* parameter. These findings provide preliminary support for the subsequent analysis of the results.

### 3.2. Mediation Analyses

Next, we examined whether alexithymia mediated the relationships between primary and secondary psychopathy and deontological and utilitarian response tendencies. The bootstrap procedure with 5000 replicates and a 95% confidence interval was adopted to examine the significance levels of the indirect effect model, with age and gender as control variables. The tested indirect effect model obtained acceptable fit indices (*χ*^2^/*df* = 4.82, RMSEA = 0.06, CFI = 0.95, TLI = 0.93, SRMR = 0.05), and all the factor loadings were highly significant (*p* < 0.001) (see Figure 1).

However, our hypothesis was only partially verified, as the mediation analysis results show that alexithymia (95% CI = [−0.135, −0.024]) only played a significant and independent mediating role between secondary psychopathy and deontological response tendencies (see Table 2). Specifically, secondary psychopathy had a highly significant positive effect on alexithymia (β = 0.61, *p* < 0.001), which in turn had a significant negative effect on deontological response tendencies (β = −0.12, *p* < 0.01).

### 3.3. Gender Difference

Firstly, this study tested whether there were gender differences in the five variables. The results indicated that the gender differences in secondary psychopathy [*t* (1168) = 0.29, *p* = 0.772], alexithymia [*t* (1168) = 0.15, *p* = 0.877], and deontological tendencies [*t* (1168) = 0.29, *p* = 0.775] were not statistically significant. However, the gender differences in primary psychopathy [*t* (1168) = 3.89, *p* < 0.001] and utilitarian tendencies [*t* (1168) = 2.80, *p* = 0.005] were significant, with males scoring higher than females on primary psychopathy, and females scoring higher than males on utilitarian tendencies.

Because this study found gender differences, multi-group analyses was conducted to examine whether the indirect effect model differed on the basis of gender. Following the suggestion of Byrne (2001), two models that keep the basic parameters equal were built [50]. The first model allows free estimation of path coefficients between males and females (unconstrained structural paths), while the second model constrains all path coefficients to be equal (constrained structural paths). Both of the models had good fitness (unconstrained structural paths: *χ*^2^/*df* = 2.63, RMSEA = 0.04, CFI = 0.97, TLI = 0.95, SRMR = 0.05; constrained structural paths: *χ*^2^/*df* = 2.44, RMSEA = 0.04, CFI = 0.96, TLI = 0.96, SRMR = 0.05). For the indirect effect model for females and males, see the Appendix A.

The results show that the chi-square differences between the two models were not significant, Δ*χ*^2^= 21.95, *p* = 0.145, suggesting that the indirect effect model did not differ by gender. This study further calculated the critical ratios of differences (CRD) to assess the between-group differences in each path coefficient [51]. If the absolute value of the CRD is greater than 1.96, it indicates a significant difference between the two parameters. The results show that none of the path coefficients were significantly different (CRD_PP→A_ = 0.089, CRD_PP→*D*_ = 0.221, CRD_SS→A_ = 1.009, CRD_SS→*D*_ = 0.588, CRD_SS→*U*_ = 0.29, CRD_A→*D*_ = 0.26, CRD_PP→*U*_ = 1.154, CRD_A→*U*_ = −0.507), suggesting that there was no difference between females and males in the indirect effects model.

## 4. Discussion

The purpose of this study is to provide more nuanced insights into the effects of sub-dimensional levels of psychopathy on moral dilemma judgments. To this end, this study examined the effects of primary and secondary psychopathy on utilitarian and deontological response tendencies. Moreover, this study also explored the mediating role of alexithymia as well as the moderating role of gender in these effects. Overall, the findings partially support the hypotheses of the present study.

As hypothesized, the study found that primary psychopathy was negatively correlated with deontological response tendencies and uncorrelated with utilitarian response tendencies. This finding suggests that individuals with high primary psychopathy are less likely to reject harm across moral dilemmas. Previous studies have shown that individuals with high primary psychopathy exhibit a preference for utilitarian judgment [24,28], and the current research found similar results (see the Appendix A). In terms of the traditional approach (i.e., deontological judgment is treated as the pure inverse of utilitarian judgment), this finding would be interpreted as the counterintuitive conclusion that people with high primary psychopathy are more likely to maximize overall well-being. The finding of this study resolves this paradox, by showing that people with high primary psychopathy are less likely to reject harm, rather than more likely to maximize overall outcomes in moral dilemmas.

This study also found that secondary psychopathy correlated negatively with deontological response tendencies. However, contrary to the expected assumption, secondary psychopathy correlated positively with utilitarian response tendencies. This finding suggests that individuals with high secondary psychopathy are less likely to reject harm and more likely to maximize outcomes across moral dilemmas. Previous studies have shown that secondary psychopathy correlated positively with utilitarian judgment [22,26], and a similar result was found in this study (see the Appendix A). The current findings provide an explanation for this result, by showing that people with high secondary psychopathy are less likely to reject harm and more likely to maximize outcomes in moral dilemmas. Although the relationship between secondary psychopathy and utilitarian response tendencies is contrary to the hypotheses of the present study, it is consistent with previous studies. Luke et al. (2022) found a marginally significant positive correlation between lifestyle–antisocial traits (similar to secondary psychopathy) and sensitivity to consequences (similar to utilitarian response tendencies), suggesting that people with lifestyle–antisocial traits are more likely to maximize outcomes in moral dilemmas [52].

Moreover, the current study hypothesized that alexithymia would mediate the relationships between primary and secondary psychopathy and deontological response tendencies. However, this study found that alexithymia only mediated the relationship between secondary psychopathy and deontological response tendencies. This finding reveals that people with high secondary psychopathy have high alexithymia, which in turn decreases deontological response tendencies. Several studies provide support for our findings. For example, Lander et al. (2012) revealed that secondary psychopathy, but not primary psychopathy, predicted a higher level of alexithymia, and they argued that this finding indicates that core emotional deficits appear to be unique to secondary psychopathy [44]. Ridings and Lutz-Zois (2014) further demonstrated that emotional processing deficits could explain why individuals with high secondary psychopathy have high alexithymia [8]. In addition, previous research suggests that utilitarian response tendencies are mainly correlated with deliberative cognitive processing of consequences, whereas deontological responses tendencies are mainly related to affective processing of harmful actions [14,29]. Alexithymia is characterized by difficulty in identifying and describing one’s own feelings [36]. Difficulties in correctly identifying one’s own feelings may lead to difficulties in identifying the emotional and mental states of others [53]. For example, individuals with high alexithymia have less empathic concern for victims in moral dilemmas [31,37]. Thus, alexithymia uniquely predicts deontological response tendencies, rather than utilitarian response tendencies. This is consistent with previous studies. Zhang et al. (2020) demonstrated that alexithymia was negatively correlated with deontological response tendencies and uncorrelated with utilitarian response tendencies [31]. Furthermore, this study also examined the moderating role of gender in these effects, showing that males scored higher than females on primary psychopathy, while females scored higher than males on utilitarian tendencies. These findings indicate that males have more interpersonal and affective deficits than females, similar to the findings of previous studies [39,54]. These findings also indicate that females are more concerned with maximizing good consequences in moral dilemmas. However, the results of the multi-group analysis show that none of the path coefficients were significantly different, suggesting that there was no difference between females and males in the indirect effect model.

This study benefited from separately quantified deontological and utilitarian response tendencies among individuals with primary and secondary psychopathy. This study found that primary and secondary psychopathy have a similar effect on deontological response tendencies, but the mechanism of such a relationship is different. Moreover, primary and secondary psychopathy have different effects on utilitarian response tendencies. These findings suggest that it is necessary to consider the separate effects of primary and secondary psychopathy on moral dilemma judgments. More importantly, the present study also has some practical implications. This study found that alexithymia not only mediated the relationship between secondary psychopathy and deontological response tendencies but also mediated the relationship between psychopathy and deontological response tendencies (see the Appendix A). These findings suggest that people with high psychopathy have high alexithymia, which leads them to be less concerned about avoiding harm. In other words, increased alexithymia leads to atypical moral judgments in psychopathy. This suggests that future treatment programs for psychopathic individuals should try to reduce alexithymia [4].

## 5. Limitations

This study also has several limitations. First, although hypothetical dilemmas can measure individuals’ moral concern, the possibility remains that decisions on real dilemmas may differ [32]. Future research could update some real dilemmas, such as individuals or medical workers infected with COVID-19, and measure individuals’ moral concern in such real dilemmas. Second, this study found that alexithymia did not mediate the relationship between psychopathy and utilitarian response tendencies. Future research should examine the possible psychological and neural mechanisms underlying this relationship. Finally, the participants in this study were undergraduate students, which may affect the representativeness of the results. Thus, future research should consider whether the present findings are applicable to other samples.

## 6. Conclusions

This study provides more nuanced insights into the relationship between primary and secondary psychopathy and moral dilemma judgment. People with high primary psychopathy are less likely to reject harm in moral dilemmas. By contrast, people with high secondary psychopathy have high alexithymia, which leads them to be less concerned about avoiding harm; in addition, they are more likely to maximize outcomes in moral dilemmas. Our findings shed new light on the moral dilemma judgment of individuals with primary and secondary psychopathy.

## Figures and Tables

**Figure 1 healthcare-10-01650-f001:**
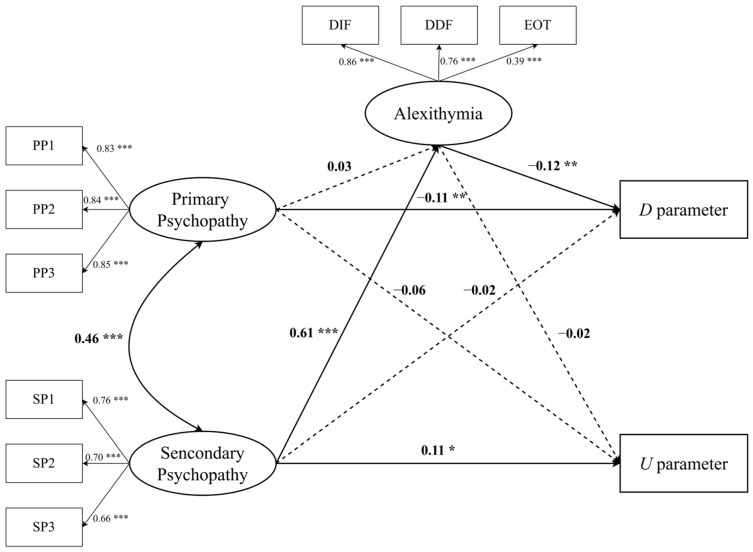
Mediation model from primary and secondary psychopathy to deontological and utilitarian response tendencies. The standardized coefficients are presented above the arrow. PP1, PP2, and PP3 are parcels of primary psychopathy. SP1, SP2, and SP3 are parcels of secondary psychopathy. DIF, DDF, and EOT are dimensions of alexithymia. * *p* < 0.05; ** *p* < 0.01; *** *p* < 0.001.

**Table 1 healthcare-10-01650-t001:** Descriptive and Pearson correlations among the variables (*N* = 1170).

Variables	*M*	*SD*	1	2	3	4
1. *U* parameter	0.14	0.23	1			
2. *D* parameter	0.63	0.23	0.11 ***	1		
3. Primary psychopathy	30.87	5.84	−0.03	−0.15 ***	1	
4. Secondary psychopathy	21.83	3.29	0.06 *	−0.12 ***	0.38 ***	1
5. Alexithymia	49.88	9.64	0.01	−0.16 ***	0.29 ***	0.51 ***

Note: *U* parameter = utilitarian response tendencies; *D* parameter = deontological response tendencies; *M* = mean; *SD* = standard deviation; ** p* < 0.05; **** p* < 0.001.

**Table 2 healthcare-10-01650-t002:** Standardized indirect effects and 95% confidence intervals.

Pathways	Estimate	Lower	Upper
1. PP→A→*D*	−0.004	−0.019	0.005
2. PP→A→*U*	−0.001	−0.011	0.002
3. SP→A→*D*	−0.074	−0.135	−0.024
4. SP→A→*U*	−0.013	−0.070	0.043

Note: PP = primary psychopathy; SP = secondary psychopathy; A = alexithymia; *U* = *U* parameter (utilitarian response tendencies); *D* = *D* parameter (deontological response tendencies).

## Data Availability

The datasets analyzed in this study are available from the corresponding author on reasonable request.

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
