# Peer review of "Effects of Primary and Secondary Psychopathy on Deontological and Utilitarian Response Tendencies: The Mediator Role of Alexithymia"

_healthcare, 2022, doi:10.3390/healthcare10091650_

Round 1

Reviewer 1 Report

Dear Authors, 

Thanks a lot for an interesting paper that may increase the readability of the journal. The following are merits of the paper:

1. The title is concise and informative; it is often used in information-retrieval systems.

2. The abstract is concise and factual, it briefly stated the purpose, the principal results, and major conclusions of the paper. 

3. Methods of the study are clear and display a clear idea of the study procedures as it made the paper be replicable, repeatable, and robust.  

4. Results of the study have been described in simple terms what the data show, it has made reference to statistical analyses used in the study. They also evaluated the trends observed and explained the significance of the results to wider understanding. 

Yet, there are minor drawbacks that should be eradicated for better paper quality:

- There should be equivalence and balance in displaying study measures. 

- Due language editing and proofreading is necessary as there are linguistics flaws (i.e., callous lack empathy, line 70; Individuals with high psychopathy are lack of emotional awareness and shallow emotions, line 135).

Reviewer 2 Report

I reviewed your work on "Effects of primary and secondary psychopathy on deontological and utilitarian responses: The role of alexithymia." Although your work is important on medical ethics, I found important problems in your work, and I have listed them one by one below.

- Add the mediator as “The mediator role of alexithymia” to the article title.

- The first sentence of the summary is not written in full. Please complete the other half.

- Make the second sentence of the summary two separate sentences. Otherwise, it got very confused and intertwined.

-The methods part is missing in the abstract. It is not clear by what method the data was collected. Also, write which analyzes were used and similar information.

- “Imagine a doctor working in your hospital became infected with a rare virus” is not an academic language and style used in the introduction. Attention should be paid to this academic language.

- The introduction should be divided into certain sub-headings and rearranged accordingly.

- Direct and indirect hypotheses or research questions should be written under the relevant subheadings in the introduction, and these hypotheses should be evaluated in the discussion section.

- The study should be based on a theory, but no such theory has been written.

- In the Method section, it should be written under the Data Analysis title or another title, which programs are used and which analyzes and data are analyzed. The process should be specified in detail.

-Limitation part should be written under a separate heading.

Reviewer 3 Report

This is an interesting study, which potentially sheds some new light on the investigation of the relationship between psychopathy and moral reasoning. The so called process dissociation approach is elegant and intuitive.

I have only a few minor suggestions which need to be addressed:

Please add the descriptive statistics for each group of the dilemmas (not just the U and D parameters, but the means, variances and ranges of responses across groups, or better still, for each dilemma separately: how many participants gave the utilitarian vs. the deontological response for each dilemma?)

I would strongly suggest the authors to test the possibility that gender also plays (a potentially moderating) role here. Gender differences have previously been shown I both alexithymia and psychopathy. Furthermore, results of some studies also suggest that there is a difference in the strength of utilitarian tendencies between genders.

To that matter, I believe the introduction might also benefit from a wider overview of individual differences (such as gender, age, education and culture) which predict utilitarian vs. deontological moral reasoning.

I appreciate the fact that the authors provided moral dilemmas in the supplementary materials. I would appreciate it even more if they were translated into English, so that interested researchers from the non-Chinese speaking background might replicate the study. Especially given the content-sensitive nature of such dilemmas, it is important to make them intelligible for all readers.

Round 2

Reviewer 2 Report

Everything I wanted for the work was done.

thanks.

Author Response

Dear reviewer,

Thank you very much for your approval of the revised manuscript.